# Exploring the clinical utility of rhythmic digital markers for schizophrenia

Axel Constant[1]*, Vincent Paquin[2,3], Robert A. Ackerman[4], Colin A. Depp[5], Raeanne C. Moore[5], Philip D. Harvey[6], Amy E. Pinkham[4]

**1** Department of Engineering and Informatics, The University of Sussex, Brighton, United Kingdom, **2** Lady Davis Institute for Medical Research, Jewish General Hospital, Montreal, Quebec, Canada, **3** Department of Psychiatry, McGill University, Montreal, Quebec, Canada, **4** Department of Psychology, School of Behavior and Brain Sciences, The University of Texas at Dallas, Richardson, Texas, United States of America, **5** Department of Psychiatry, School of Medicine, University of California San Diego, La Jolla, California, United States of America, **6** Department of Psychiatry and Behavioral Sciences, University of Miami, Miami, Florida, United States of America

☺ These authors contributed equally (co-first authors).
* axel.constant.pruvost@gmail.com

## Abstract

This study investigates the clinical utility of rhythmic digital markers (RDMs) in schizophrenia. RDMs are digital markers capturing behavioral rhythms over different timescales - within 24 hours span (ultradian), at a span of 24 hours (circadian), or over cycles of more than 24 hours (infradian). While previous research has explored digital markers for schizophrenia, the focus has primarily been on sensor data variability rather than rhythmic patterns. This study introduces two RDMs: an entropy RDM, which quantifies uncertainty in activity distribution over the infradian cycles, and a dynamic RDM, which is derived from models of transitions in entropy and psychotic symptom intensity using Markov chain analysis. Data were ecological momentary assessments (EMAs) of 39 activities collected from 390 individuals diagnosed with schizophrenia (N = 153) or bipolar disorder (N = 192) and controls (N = 45). We assessed associations between RDMs and symptom severity and whether participants could be differentiated based on these RDMs. We found that participants with schizophrenia significantly differed on dynamic RDMs, suggesting a potential diagnostic utility. However, dynamic RDMs were not associated with symptom severity, and entropy RDM had no significant clinical correlate. Our findings contribute to the growing evidence on digital markers in psychiatry and highlight the potential of rhythmic digital markers (RDMs) in characterizing digital phenotypes for schizophrenia.

## Author summary

In our study, we explored new ways to track schizophrenia using digital tools that measure daily activity patterns. These tools, called rhythmic digital markers

**Data availability statement:** Data are publicly available on the National Institute of Mental Health Data Archive (NDA) website through following the instruction for access and searching for the dataset titled "Introspective Accuracy, Bias, and Everyday Functioning in Severe Mental Illness" (Collection ID: C2941). The link is: https://nda.nih.gov/general-query.html?q=query=data-structure%20~and~%20orderBy=shortlabel%20and~%20orderDirection=Ascending.

**Funding:** AC is supported by a European Research Council Grant (XSCAPE) ERC-2020-SyG 951631. The original data featured in this study was supported by NIMH grant RO1MH112620 to AEP. VP is supported by an award from the Fonds de recherche du Québec - Santé and Ministère de la santé et des services sociaux (FRQS-MSSS). The funders had no role in study design, data collection and analysis, decision to publish, or preparation of the manuscript.

(RDMs), analyze behavior over different time periods—short cycles within a day, full-day rhythms, and patterns lasting multiple days. Our study introduces two new RDMs: one that measures the unpredictability of activity patterns over longer cycles and another that tracks how changes in activity relate to psychotic symptoms. To test these markers, we collected data from 390 individuals, including those with schizophrenia, bipolar disorder, and healthy participants. Our results showed that people with schizophrenia had distinct rhythmic patterns, suggesting that these markers could be useful for diagnosis. However, we found that they did not strongly relate to symptom severity. Our findings add to the growing research on digital technology for psychiatry and suggest that rhythmic digital markers could play a role in identifying schizophrenia in the future.

# 1 Introduction

## 1.1 The clinical utility of digital markers

This study explores the clinical correlates of Rhythmic Digital Markers (RDMs) in schizophrenia, based on the frameworks of precision psychiatry, information theory, and dynamical systems theory. Precision psychiatry aims at the "tailoring of medical treatment to the individual characteristics of each patient" and at the "classification of individuals into subpopulations that differ in their susceptibility to a particular disease, in the biology and/or prognosis of those diseases they may develop, or in their response to a specific treatment" [1]. This is achieved by identifying biomarkers that have clinical utility [2], that is, that are useful with respect to the different clinical factors (e.g., prognostic, diagnostic, therapeutic, nosological, monitoring, etc.). Although biomarkers derived from low-level biological processes have had limited success in psychiatry [3–5], there remains excitement around markers configured at higher levels such as digital biomarkers derived directly from data sampled using digital technologies [6]. Because there is a controversy on the use of the term biomarker to qualify markers that do not derive from low-level biological processes [7], here we will refer to digital biomarkers as "digital markers".

A marker, digital or otherwise, has diagnostic utility if it indicates the presence of an illness through diagnostic testing and has therapeutic utility if it can function indirectly as a treatment target or indicator of treatment success [8]. A marker has prognostic utility if it predicts the course of an untreated disease, and predictive utility if it predicts the course with treatment [9]. A marker has nosological utility (i.e., it can ground the definition of a disease or disease subtype) if it cumulates all the aforementioned utilities. The first step in achieving clinical utility for a given marker is to find an association between the marker and the utility target (e.g., association with symptom course in case of prognostic utility, or differentiation of clinical and healthy participants in case of diagnostic utility). Following the identification of such correlates, the demonstration of clinical utility requires benchmarks such as specificity and sensitivity, which need to be validated in independent samples [10].

Digital markers with clinical utility can be derived from actively sampled ecological momentary assessment (EMA) data [11] by querying users throughout the day. They can also be passively sampled via mobile sensing data from wearable or portable devices such as smartphones and smart watches [12,13]. Examples of prognostic and diagnostic digital markers for schizophrenia include markers derived from passive sensing data (e.g., location, sleep patterns, calls, SMS text messages) sampled through smartphones [14–16], vocal signatures [17], facial characteristics (e.g., facial expressivity) [18] (for reviews see [19–21]), and motor behavior such as the rate of head movement measured using computer vision software and sampled through daily remote video recording with smartphones [22].

## 1.2 Rhythmic digital markers

Because it is sampled periodically throughout the day, EMA data reflects information that pertains to the unfolding over time of cycles that happen at different time scales, within 24 hours span (ultradian), at a span of 24 hours (circadian), and over cycles of more than 24 hours (infradian). Such rhythms help in adapting to the structure of a changing environment and are known to influence mental health [23,24]. Rhythmic information captured through EMA tracks social rhythms, which refer to common social and lifestyle activity patterns that exert effects on underlying biological rhythms and contribute to anchoring a person's social interaction (e.g., bedtimes, mealtime locations, etc.) [25>,26].

The clinical utility of social rhythms has been studied primarily for bipolar disorders [27], and social rhythms are thought to relate to depression and anxiety [28]. But some have argued for their therapeutic significance in the context of schizophrenia [29]. Here we call rhythmic digital markers (RDM) the digital markers derived from an analysis of rhythmic information captured by digital data such as EMA and passive sensing data. Several previous works align with the concept of RDMs. For instance, behavioral routine information collected through passive sensing data was shown to be associated in clinical participants with self-reported anxiety, depression, psychosis and sleep quality [30]. Symptom changes and time spent at home measured using smartphone passive and active data over 6 months were found to be highly correlated with clinical assessment scores in schizophrenia patients [31]. Changes in mobility and time spent at a participant's primary location measured through geolocation data have been used to predict psychosis relapse [32, 33], and the entropy of geolocation data along with the time spent at the participants' primary location as well as the distance between locations and distance travelled has been predictive of positive symptom scores [34].

RDMs thus have potential utility as diagnostic and prognostic markers in schizophrenia. However, research in this area to date has been limited to the variability of sensor data (e.g., geolocation) or specific categories of symptoms or behaviors, using indices of variability such as entropy, a concept drawn from Shannon's information theory [35]. While useful, indices of variability do not capture time-dependent patterns in the fluctuations of symptoms or behaviors over time. Dynamical systems theory of mental illness suggests that mental health problems can be characterized according to their transitions over time between states, with a tendency of some states to "pull" and "push" individuals into and away from them for various periods of time, thereby explaining characteristic patterns in mental disorders configured at different levels of organizations, from the neurobiological, to the behavioral and social interactional level [36]. Drawing from the idea of social rhythms and dynamic systems theory, we propose to explore the clinical correlates of RDMs based on *time-dependent patterns* of activities entropy and self-reported psychotic symptoms. We use Markov chains to build dynamic indices – the RDMs – out of the fluctuations in those otherwise static indices.

## 1.3 This study

We aimed to begin the exploration of the prognostic and diagnostic utility of RDMs derived from EMA data about social rhythms (cf.[37]). This study was conducted in a sample of adults diagnosed with schizophrenia or bipolar disorder and controls. We used the information theoretic construct of Shannon entropy and Markov chain modelling to build dependent variables out of EMA data on activities forming social rhythms. We explored the potential for diagnostic and prognostic

utilities of two variables, or RDMs. The first variable—the entropy, or static RDM—is the Shannon entropy (H) of the distribution of activities over infradian activity cycles. The second variable—the dynamic RDM—measures the dynamic properties of a Markov chain model of the changes, or transitions in the first variable (i.e., change in entropy) along with change in psychotic symptoms intensity. We explored the clinical correlates of these RDM by testing their associations with symptom severity and their differences between diagnostic groups (schizophrenia, bipolar disorder, and control).

**1.3.1 Entropy modelling for information theoretic RDM.** Modelling of Shannon entropy and Markov chains was done in MATLAB version R2024b with the econometrics toolbox. The Shannon entropy (H) can be measured for a probability distribution using:

$$H(X) = -\sum_{x \in X} p(x) \log(p(x))$$

(1)

In eq. 1, H(X) represents the H of a random variable X with elements x forming a probability distribution. Formally, H represents the level of uncertainty over X. To make the notion of Shannon entropy (H) more intuitive, it is useful to view it as capturing the level of certainty or credence that an observer should have when sampling X at random. For instance, a distribution made of two elements representing "heads" and "tail" would attribute, for a well-balanced coin, 50% probability to each element of the distribution, or side of the coin. This distribution could be obtained, for instance, by recording the frequencies of heads and tails over, say, a million coin flips. H for that distribution would be 1 (i.e., full uncertainty), which is the maximum entropy for a distribution with 2 elements. This means that the level of credence one should have with respect to winning versus losing a bet on a coin flip should be minimal; indeed, a maximum entropy corresponds to a minimal level of credence, reflecting the inherent unpredictability of the coin flip. The RDMs we develop in the present study are built out of the Shannon entropy of participants' social rhythms. This means that they track the unpredictability of the activity patterns of participants in each group (e.g., schizophrenia, bipolar, and healthy controls), while reflecting the level of credence one should have over the possibility of finding a participant engaged in one of the 39 possible activities if sampled at random (e.g., how confident a clinician should be that they could predict the activity pattern of their new client vs. in healthy populations).

To capture the Shannon entropy of social rhythms, we combined EMAs into cycles of 3 days (i.e., 9 EMAs per cycle), for a maximum of 10 cycles over the study period. We represented the frequencies per participant of each activity per cycle of 3 days as a probability distribution. The number of elements in the distribution corresponded to the number of distinct activities done in the 3-day period. If, say, "watching TV" recurred 5 times over the 3-day period, and there were only 2 activities (e.g., "watching TV" and "cooking") and the second activity recurred 4 times, then the distribution would have 2 elements; watching TV (P(X) = [watching TV 5/9; cooking 4/9]). In this case, the distribution over X would yield an entropy of .991 (on a maximum possibility of 1), which is high, and would indicate greater uncertainty as to the outcome of sampling at random of the activity distribution of this cycle (see the bar plot in Fig 1). H was only calculated for cycles with fewer than 3 of 9 missing observations; in these cases, missing observations were treated as a distinct activity.

**1.3.2 Markov chain modelling for RDMs.** Markov chains are part of the family of Markovian models [38], which are statistical models used to model autonomous (e.g., chemical processes) and controlled (e.g., decision making processes) processes that involve conditionally independent events. Conditionally independent events are events whose probability of occurrence only depends on a related past event (e.g., an event at time t being conditioned on an event at time t-1, but not on time t-2...t-n). Markov chains are interesting because they allow us to uncover dynamic properties of time series data such as the Expected Hit Times (EHT) of states modelled by the Markov chain, or the number of steps the chain is expected to take before arriving at a given state, when starting from any given state. Here, we use a Markov chain to model the processes of combined increases and decreases in cycle activity entropy (H) and mean self-reported symptoms (SC) over time in each participant. We thus assume that reported symptoms (SC) and change in activity entropy (H) both result from an autonomous process. The state space for the Markov chain is made of 9 states combining possible H data and SC delta (Fig 1 and Box 1).

PLOS Digital Health

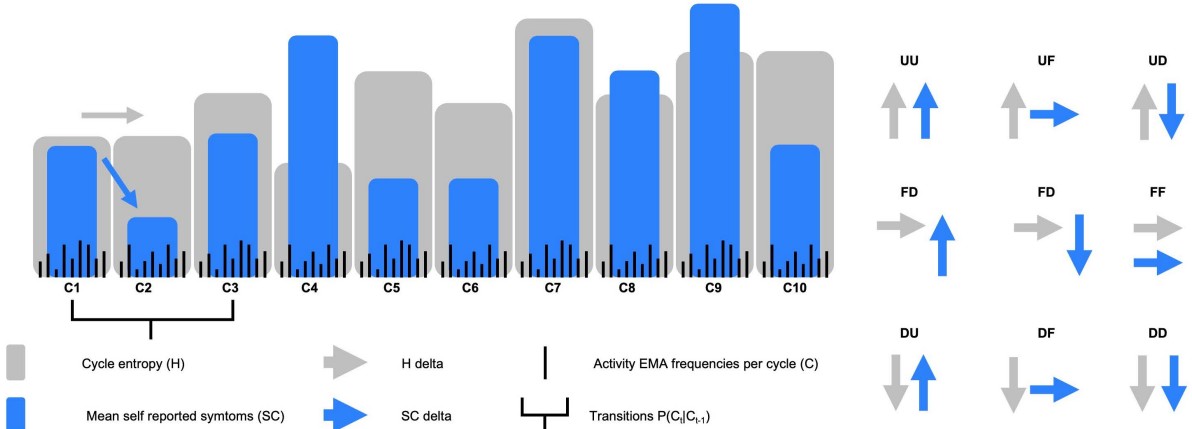

**Fig 1. Left hand side. "Each gray bar represents entropy (H) of the distribution of activities sampled over 3 days (up to 9 black bars representing EMAs frequencies).** Each blue bar indicates the mean self-reported symptom score over the 3 days period (SC). The bar plot illustrates possible values of H and SC of a hypothetical participant over 10 three days periods/ cycles (C1-C10). Right hand side. The delta in H (grey arrow) and SC (blue arrow) were combined to create the 9 possible states for the transition matrix of the Markov chains.

---

**Box 1. States of the Markov chains.**

1. H up & SC up (UU)
2. H up & SC flat (UF)
3. H up & SC down (UD)
4. H flat & SC up (FU)
5. H flat & SC flat (FF)
6. H flat & SC down (FD)
7. H down & SC up (DU)
8. H down & SC flat (DF)
9. H down & SC down (DD)

H = cycle entropy; SC = self-reported psychotic symptoms per cycle

---

The probabilities that make up the Markov chain are denoted as:

$$p_{ij} = P(X_t = j | X_{t-1} = i)$$

(2)

where $p_{ij}$ is the probability of transitioning from a state of the random variable X at t-1 to another state of X at time t. These probabilities are encoded in a transition probability matrix P with rows representing states at time t-1 and columns representing states at time t; the rows of the matrix sum to 1 to form a probability distribution.

We obtain the probabilities for the Markov chain by counting the number of times each possible transition between all possible states (81 possible transitions) have been engaged by a participant. This produces an empirical probability transition matrix that can be normalized to ensure that each row sums to 1. We attribute a small probability to all the transitions in the matrices of all the participants ($p_{ij} = 0.1$), clinical and control. This ensures that after normalization all the rows contain at least some probability. We obtained a transition probability matrix for each participant, which we used to compute the mean value of the expected hit times (EHT) from each state to each of the 9 states. The modelling of Markov chains

required at least 3 cycles with available data to observe at least 2 states per participant. This means that to be included in the analyses, participants' data had to present (i) fewer than 3 missing observations per cycle (ii) over at least 3 cycles. We compute the expected hit time to go from any state to a state of interest using the hittime function of the MATLAB econometrics Toolbox.

We also computed the total entropy of the transition matrix of each participant (MH) by summing the Shannon entropy for each row distribution. Finally, using the aggregated transition probability matrices for all participant's Markov chain, for each group, we simulated a random walk, which is a stochastic process describing the path taken by a system when randomly progressing through its state space. Here, the system is the combined H and SC fluctuations over the 30 days. The random walk was generated by, first, randomly sampling a number in a [0 1] interval, and then by running the cumulative sum of the row distribution of the transition matrix, starting at state 1, and picking the state that corresponds to the interval of the cumulative sum matching the random number. For instance, if a distribution is [.2.4.4], its cumulative sum is [.2.6 1], and if a random number is, say,.56, then the selected state is state 2, since.56 is between.6 and.2. We ran the random walk 100,000 times, restarting each time with the distribution corresponding with the state previously selected.

## 2. Results

### 2.1 Sample characteristics

Of 500 participants recruited in the original sample, 390 participants met the minimum requirement of 3 cycles (i.e., 9 days) of data for inclusion in the analyses (see Table 1 for a description of the sample). Compared to participants included in the analyses, those removed (n = 110) were in greater proportion in the schizophrenia, Hispanic, White, and Other ethnoracial groups, and had higher levels of clinician-reported positive symptoms (S1 Table).

### 2.2 Associations of H with self-reported psychotic symptoms

Person-level mean of H was not significantly associated with momentary psychotic symptoms (mean ratio = 1.013; 95% CI: 0.891, 1.152). Mean-centered H was also not significantly associated with these symptoms (mean ratio = 1.001; 95% CI: 0.979, 1.023). Associations were also not significant in analyses restricted to the schizophrenia group (person-level mean of H, mean ratio = 1.103; 95% CI: 0.892, 1.364; mean-centered H, mean ratio = 0.985; 95% CI: 0.943, 1.028), or in reverse models where H was regressed on person-level mean symptoms (regression coefficient = 0.000; 95% CI: -0.010, 0.011) and mean-centered symptoms (regression coefficient = 0.001; 95% CI: -0.020, 0.017).

### 2.3 Associations between RDMs and interviewer-rated symptoms

The RDMs were not significantly associated with interviewer-rated symptom severity after correction for false discovery rate (Fig 2).

### 2.4 Group differences in RDMs

Group differences in RDMs are shown in Fig 3, and here we report those that were significant after correction for false discovery rate. Relative to the control group, participants with bipolar disorder did not differ on any marker. Relative to the control group, participants with schizophrenia had shorter Mean Expected Hit Times (MEHTs, shown as Mean hit time in Figs 2 and 3) for UU, UD, DU, and DD, and longer MEHT for UF and DF. They did not significantly differ on other markers (mean H, MH, and other MEHTs). Relative to participants with bipolar disorder, those with schizophrenia had shorter MEHTs for UD, DU, and DD, and longer MEHTs for UF and DF.

### 2.5 Markov chain simulation

The simulation results in Fig 4 show the distribution of states visited over 100,000 random walks in the state space of the aggregated Markov chains of controls and schizophrenia participants. The entropy H for each distribution is indicated next

**Table 1. Description of the sample.**

| | Control N=45 | Bipolar disorder N=192 | Schizophrenia N=153 |
|---|---|---|---|
| Suite of recruitment, N (%): | | | |
| The University of Texas at Dallas | 45 (100%) | 76 (39.6%) | 71 (46.4%) |
| University of Miami | 0 (0.00%) | 50 (26.0%) | 50 (32.7%) |
| University of California San Diego | 0 (0.00%) | 66 (34.4%) | 32 (20.9%) |
| Age in years, Median [25th;75th] | 36.0 [32.0;46.0] | 39.0 [30.0;50.0] | 41.0 [33.0;53.0] |
| Gender, N (%): | | | |
| Men | 23 (51.1%) | 56 (29.2%) | 77 (50.3%) |
| Women | 22 (48.9%) | 135 (70.3%) | 76 (49.7%) |
| Other | 0 (0.00%) | 1 (0.52%) | 0 (0.00%) |
| Ethnoracial groups, N (%): | | | |
| Asian | 5 (11.1%) | 14 (7.29%) | 5 (3.27%) |
| Black | 12 (26.7%) | 38 (19.8%) | 80 (52.3%) |
| Hispanic | 10 (22.2%) | 47 (24.5%) | 29 (19.0%) |
| Other | 1 (2.22%) | 11 (5.73%) | 8 (5.23%) |
| White | 17 (37.8%) | 82 (42.7%) | 31 (20.3%) |
| Educational attainment, N (%): | | | |
| High school diploma or less | 14 (31.1%) | 36 (18.8%) | 70 (45.8%) |
| Some college | 14 (31.1%) | 62 (32.3%) | 58 (37.9%) |
| College degree or higher | 17 (37.8%) | 94 (49.0%) | 25 (16.3%) |
| Relationship status, N (%): | | | |
| Not in a relationship | 21 (46.7%) | 87 (45.3%) | 95 (62.1%) |
| In a relationship | 24 (53.3%) | 105 (54.7%) | 58 (37.9%) |
| Positive psychotic symptoms, Median [25th;75th] | NA | 11.0 [9.00;13.0] | 16.0 [13.8;19.0] |
| Reduced emotional experience, Median [25th;75th] | NA | 3.00 [3.00;6.00] | 6.00 [3.00;8.00] |
| Reduced emotional expression, Median [25th;75th] | NA | 4.00 [4.00;6.00] | 6.00 [4.00;8.00] |

to the title of each plot (2.3441 for the schizophrenia population, and 2.0869 for the control population). The MEHTs differentiating the schizophrenia group from the control participants are reflected in the simulation results: UF (3.7% against 13.7% hits in controls), UD (10.6% against 3.8% hits in controls), DU (25.8% against 8.8% hits in controls), DF(15.3% against 53.8% hits in controls), DD (37.8% against 14.6% hits in controls).

## 3. Discussion

### 3.1 Diagnostic utility

Our results provide preliminary evidence of the potential of RDMs as diagnostically informative digital markers (see Fig 5).

   **3.1.1 Schizophrenia vs. controls.** The associations found, and summarized in Fig 3, suggest that the schizophrenia group took significantly **more time** than controls to reach more unpredictable or less unpredictable social rhythm patterns, without noticing changes in their symptoms (UF, DF) This suggests that among participants with schizophrenia, changes in social rhythm unpredictability were more closely linked to symptom intensity than in controls, who more quickly reached states where the two were unrelated.

   Participants with schizophrenia were also expected to arrive less often at such states according to our simulation. Participants in the schizophrenia group took significantly **less time** to arrive at a state of increased unpredictability in

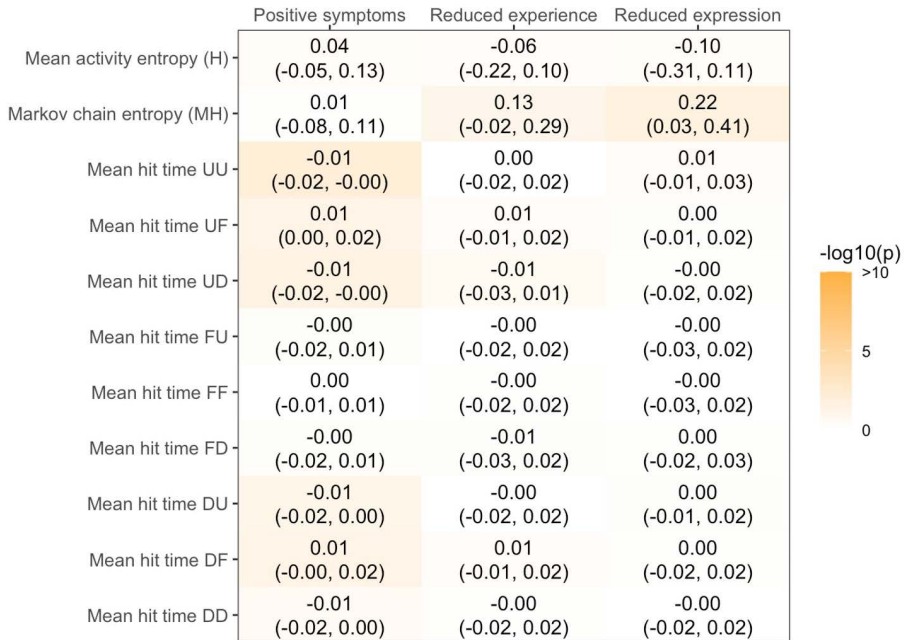

**Fig 2. Associations between RDMs and interviewer-rated symptoms.** Mean H is mean entropy over cycles, MH is the total entropy of the transition matrix, and UU, UD, UF, FU, FF, FD, DU, DF, DD are the mean hit times taken by the participants' Markov chain to reach each of those states (cf. Box 1). Estimates are regression coefficients (uncorrected 95% CI) adjusted for site of recruitment, age, gender, ethnoracial groups, educational attainment, and mean levels of self-reported psychotic symptoms. None of the associations remained significant (p < .05) after correction for false discovery rate.

social rhythms accompanied with a decrease in their self-reported symptoms, and vice versa (UD, DU). They were also expected to arrive more often at such states according to our simulation. Additionally, participants with schizophrenia took significantly less time to arrive at states of simultaneous decreased or increased unpredictability while simultaneously reporting a decrease or increase in symptoms intensity (UU, DD).

**3.1.2 Schizophrenia vs. bipolar.** The diagnostic correlates of dynamic RDMs, such as summarized in Fig 3, were similar in schizophrenia to bipolar disorder but not in bipolar disorder relative to the control group, suggesting a potential specificity to schizophrenia. Participants with schizophrenia took significantly **less time** than bipolar participants to arrive at states where they either displayed increased unpredictability of social rhythms accompanied with decreased self-reported symptoms (UD), or its opposite, that is, decreased unpredictability of social rhythms and increased in self-reported symptoms (DU), or to a state where they displayed decreased unpredictability and decreased self-reported symptoms (DD). Participants with schizophrenia also took significantly **more time than** participants with bipolar disorder to arrive either at states where they displayed increased unpredictability of social rhythms and steady self-reported symptoms (UF), or at states where they displayed decreased unpredictability of social rhythms and steady self-reported symptoms intensity (DF).

**3.1.3 Interpretation of findings.** Our findings suggest a dynamic systems feature associated with schizophrenia, whereby the entropy (i.e., unpredictability) of activities tends to fluctuate more when there are concurrent fluctuations in psychotic symptoms. This feature was observed at the group level, i.e., when comparing schizophrenia to bipolar disorder and control groups, and concerns the *variability* of entropy, which should be contrasted with the above negative finding concerning the association of entropy itself with symptom severity. The group-level pattern is consistent with the clinical notion that the emergence of psychotic symptoms may lead to changes in functioning and routines, (e.g., yielding a greater variety and complexity of habitual activities -- higher entropy -- or the concentration of habits to a narrower set

|  | Bipolar disorder vs. control | Schizophrenia vs. control | Schizophrenia vs. bipolar disorder |
|---|---|---|---|
| Mean activity entropy (H) | 0.09 (-0.06, 0.25) | 0.00 (-0.16, 0.17) | -0.08 (-0.20, 0.03) |
| Markov chain entropy (MH) | -0.15 (-0.34, 0.05) | -0.20 (-0.41, 0.01) | -0.05 (-0.20, 0.09) |
| Mean hit time UU | -0.99 (-2.65, 0.67) | **-2.27 (-4.06, -0.47)** | -1.44 (-2.64, -0.24) |
| Mean hit time UD | -1.30 (-2.98, 0.39) | **-3.14 (-4.96, -1.32)** | **-1.97 (-3.21, -0.73)** |
| Mean hit time UF | 1.09 (-0.52, 2.70) | **3.93 (2.19, 5.68)** | **2.91 (1.70, 4.12)** |
| Mean hit time FU | 0.24 (-0.97, 1.45) | 0.71 (-0.59, 2.02) | 0.43 (-0.45, 1.32) |
| Mean hit time FF | 0.39 (-0.86, 1.65) | 1.15 (-0.20, 2.51) | 0.69 (-0.21, 1.59) |
| Mean hit time FD | -0.12 (-1.29, 1.06) | 0.53 (-0.74, 1.80) | 0.60 (-0.26, 1.46) |
| Mean hit time DU | -1.52 (-3.24, 0.19) | **-3.02 (-4.88, -1.16)** | **-1.61 (-2.87, -0.36)** |
| Mean hit time DF | 1.05 (-0.55, 2.64) | **3.73 (2.01, 5.46)** | **2.80 (1.59, 4.01)** |
| Mean hit time DD | -0.75 (-2.37, 0.87) | **-3.52 (-5.27, -1.77)** | **-2.92 (-4.10, -1.73)** |

-log10(p)
>10

**Fig 3. Group differences in RDMs.** Mean H is mean entropy over cycles, MH is the total entropy of the transition matrix, UU, UD, UF, FU, FF, FD, DU, DF, DD are the mean hit times taken by the participants' Markov chain to reach each of those states (cf. Box 1). Estimates are regression coefficients (uncorrected 95% CI) adjusted for site of recruitment, age, gender, ethnoracial groups, educational attainment, and mean levels of self-reported psychotic symptoms. Significant associations (p < .05) after correction for false discovery rate are indicated in bold. The non-standardized regression coefficient captures the group effect relative to either the control (second column) or the bipolar group (third column) on the variable described on the Y axis, with the negative values reflecting a lower (e.g., faster) value for the variable (e.g., -2.27 reflecting a shorter MEHT to UU in schizophrenia vs. controls), and with the values in bracket below indicating the confidence interval (-4.06, -0.47).

of activities -- lower entropy). Changes in activities may also have reciprocal effects on psychotic symptoms if they carry mechanisms of risk (e.g., disrupted sleep schedules) or protection (e.g., increased access to social support). The lack of association of dynamic RDMs with bipolar disorder could reflect the lower prevalence of psychotic symptoms in this population, in whom variations in activity rhythms may be more strongly related to affective symptoms such as those of depression and mania.

### 3.2 Prognostic utility

We found no significant association that suggested a prognostic utility of either the entropy, or static RDMs (i.e., mean H and MH) or dynamic RDMs (i.e., based on MEHTs). Entropy RDMs also did not suggest diagnostic utility. The above findings show that activity entropy could vary in either direction (higher or lower) in the context of psychotic symptoms. This could explain why increases or decreases in activity entropy over time did not predict psychotic symptoms within individuals. There are several reasons that may explain why entropy RDMs did not suggest prognostic or diagnostic utility in our study, despite entropy having been successfully used for other types of information and for other clinical populations [37,39,40]. The first reason might be statistical. The frequency of the sampled events (i.e., 9 samples over 3 days) was potentially too low to capture variation in activities that could be associated with variation in symptoms, particularly as self-reported symptoms had limited variability over time. For example, in a CrossCheck study (61 outpatients with schizophrenia), higher multi-scale entropy of ambient sound within periods of 3 hours collected over 6–12 days was predictive of subsequent EMA reports of being bothered by voices and worrying about people trying to harm oneself [37]. Future study should use

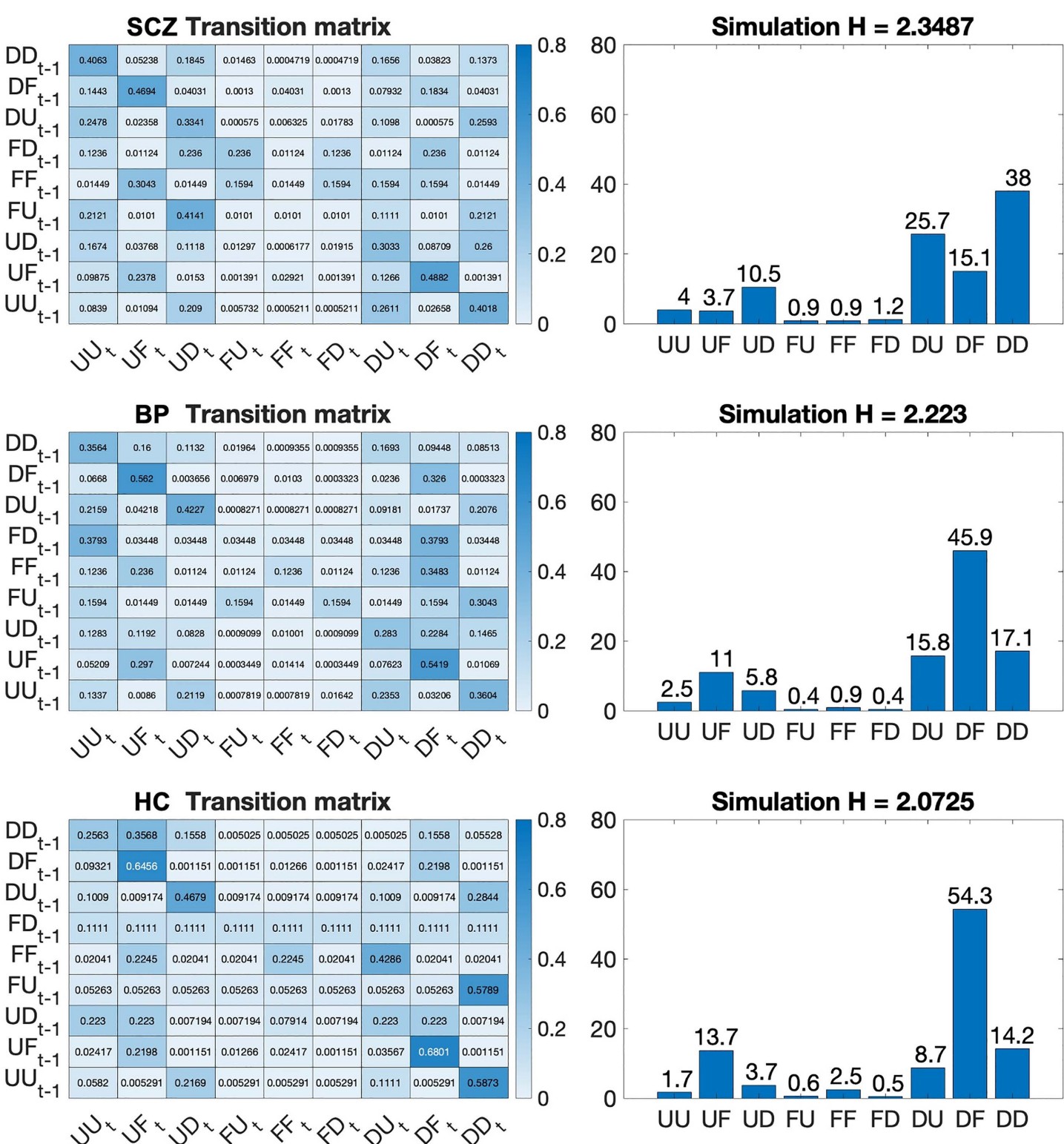

**Fig 4. (Left) Aggregated Markov chain for the schizophrenia (SCZ), bipolar (BP), and healthy control (HC) groups.** (Right) Simulation results for the aggregated Markov chain in each group. Y axes represent percentages (%) of the total states (see S1 Video for a dynamic representation of Fig 4).

| | Schizophrenia vs. Bipolar | Schizophrenia vs. Controls |
|---|---|---|
| ⬆➡ UF (Entropy Up, Symptoms Flat) | Slower MEHT (2.91) | Slower MEHT (3.93) |
| ⬇➡ DF (Entropy Down, Symptoms Flat) | Slower MEHT (2.80) | Slower MEHT (3.73) |
| ⬇⬇ DD (Entropy Down, Symptoms Down) | Faster MEHT (-2.92) | Faster MEHT (-3.52) |
| ⬇⬆ DU (Entropy Down, Symptoms Up) | Faster MEHT (-1.61) | Faster MEHT (-3.02) |
| ⬆⬇ UD (Entropy Up, Symptoms Down) | Faster MEHT (-1.97) | Faster MEHT (-3.14) |
| ⬆⬆ UU (Entropy Up, Symptoms Down) | - | Faster MEHT (-2.27) |

**Fig 5. Results summary, such as found in Fig 3.**

alternative ways of sampling information about social rhythms that allow for higher sampling frequency. A second reason may be that activity patterns are simply not features whose entropy is directly indicative of psychotic symptom trajectory; whether an individual has varied activities or not may not be so meaningful clinically as the experience, motivations, and social context of the activities, as well as lifetime disability-related factors such as social exclusion and barriers to occupational pursuits.

## 4. Conclusions

In this exploratory study, we focused on diagnoses and symptom severity as preliminary benchmarks for establishing the clinical utility of RDMs in schizophrenia. This approach was made possible by leveraging a unique dataset, with several strengths including a large clinical sample, a combination of researcher-administered and self-reported measures, and detailed activity reports over a 1-month period. We focused on infradian cycles of social rhythms by combining activity and symptom reports over epochs of 3 days. Other time periods may reveal different patterns, such as the effect of sleep schedules over circadian (24-hour) rhythms. Beyond the clinical correlates of the specific RDMs tested here, we hope that the present analytical approach can be upscaled and improved in future research to advance the development of digital markers of dynamical systems and social rhythms. However, to be successful, such efforts should center on the aspects of social rhythms and outcomes that are deemed key by people with lived experience. Established symptom or behavior scales capture only a fragment of the varied experiences of people with psychotic disorders and often do not align with what they consider most meaningful for recovery [41,42]. We therefore acknowledge the importance of grounding precision psychiatry endeavors, such as RDMs, in participatory methods and person-centered measures [43].

### 4.1 Future directions

Our method provides a new way of interpreting diagnostic information that pertains to mental disorders understood as dynamical systems, and that aligns with existing strategies for modelling traits of mental disorders based on complex systems thinking such as causal loop diagrams [44] and symptom networks [45]. The transition matrix allows demarcating dynamic features of the control and schizophrenia populations. There were fewer, though higher probability transitions in the matrix of the healthy controls. The most probable transitions were those from DF to UF and from UF to DF, which speaks to the low fluctuation in self-reported symptoms characteristic of the control population. In contrast, there was a greater number engaged transitions in the matrix representing the schizophrenia population, indicating the tendency of the system to navigate a greater variety of states. This feature was further illustrated by the simulation results, which showed the greater dispersion of the states visited by the system in the schizophrenia group, with the entropy of the distribution of simulation results in schizophrenia being 2.35, in bipolar 2.23, and in controls 2.07. The diagnostic utility of such measures remains to be confirmed through validation in independent samples, but the present results illustrate the potential for dynamical systems theory to guide the development of diagnostic digital markers. More broadly, one may consider interpreting the dynamic character of illness experience captured by social rhythms dynamics through the lens of lived experience narratives and phenomenological research, by reading (dis)continuities of social rhythms as indicators

of shifts towards the emergence or resolution of psychotic experiences. Coupled with phenomenological data, the present approach may be expanded in the future to explore how patterns of insight or realization about the self or the world [46] follow patterns of variations in social rhythms.

### 4.2 Limitations

Limitations of the study include attrition: participants who completed fewer than 3 cycles were removed from analyses, and within the analytic sample, cycles with insufficient data could not be analyzed. Attrition bias may thus affect the result, for example if rapid increases in activity entropy or psychotic symptoms interfered with participants' capacity to complete EMAs. Another limitation is the composition of the control group, which was smaller and recruited at only one of the 3 sites; these limit the statistical power and generalizability of comparisons made between the clinical and control groups. Additionally, self-reports of activities and psychotic symptoms may be affected by desirability and other forms of biases. Finally, our Markov chains were trained on a limited amount of data, given their size (9 states), which might significantly limit reproducibility of the study in the future.

## 5 Methods

### 5.1 Participants

This was a secondary analysis of an existing dataset aimed at examining insight and cognitive functioning in people with schizophrenia or bipolar disorder [47,48]. Data are publicly available on the National Institute of Mental Health Data Archive (NDA) website through following the instruction for access and searching for the dataset titled "Introspective Accuracy, Bias, and Everyday Functioning in Severe Mental Illness" (Collection ID: C2941). Participants were adults aged 18-65 years in three groups: schizophrenia (or schizoaffective disorder), bipolar disorder, and controls. Trained interviewers collected the diagnostic information using the psychosis module of the Structured Clinical Interview for the Diagnostic and Statistical Manual of Mental Disorders Fifth Edition [49]. Diagnoses were obtained based on a local consensus procedure. Included participants had to meet a clinical stability criterion (i.e., no hospitalization or extended emergency department visit for a minimum of 6 weeks) and a medication criterion (i.e., no significant (>20%) medication dose changes in the past 2 weeks). Exclusion criteria were English non-proficiency, a medical or neurological disorder affecting brain functioning such as tumors or seizures, intellectual disability or pervasive developmental disorder, active substance use (moderate severity or higher) and visual or hearing impairment that would limit smartphone use. Participants were recruited at The University of Texas at Dallas, Miller School of Medicine–University of Miami, and at the University of California San Diego. Recruitment was done in medical centers, public mental health and local community clinics, nonprofit organizations, as well as through direct contact with service providers. All participants provided written informed consent, and the institutional review board of each university approved the study.

### 5.2 Baseline assessments and EMAs

Schizophrenia-related symptom severity was assessed for clinical groups (schizophrenia or bipolar disorder) by trained raters one day before the first EMA survey. Symptoms were assessed using the PANSS [50], which rates 30 items on a scale from 1 = "Absent" to 7 = "Extreme". Consistently with previous factor analyses of the PANSS, we analyzed three scores: the positive symptom subscale (7 items; total score range 7-49), reduced emotional experience (3 items from the negative symptom subscale; total range: 3–21) and reduced emotional expression (4 items from the negative symptom subscale; total range: 4–28) [51].

EMA surveys were completed by participants via the NeuroUX platform (NeuroUX, Inc). Upon receipt of an SMS text messages prompt, participants completed the survey, using either the smartphone provided by the investigators or their own smartphone. Participants were prompted 3 times daily over 30 days, at stratified random intervals, within three predefined intervals of < 11am, 12–3pm, and 6–9pm. The times of the first and last assessment of the day were adjusted

based on the participants' sleep and wake schedules. Surveys had to be completed within a 1-hour window after the prompt, and participants could silence the smartphone alarm if needed, for 30-minute intervals.

The surveys included questions about engagement in current activities at the moment of the survey and were measured using checkbox questions on 39 possible different activities (see Box 2). Multiple activities could be checked per assessment.

Box 2. List of the 39 activities measured through ecological momentary assessments.

1. Preparing food
2. Eating or drinking
3. Cleaning my home/room
4. Laundry
5. Budgeting or paying bills
6. Showering or grooming
7. Changing clothes
8. Watching TV
9. Using social media
10. Shopping online
11. Other internet/computer/tablet use
12. Reading, Writing, or journaling
13. Gardening
14. Social interactions
15. Working (paid)
16. Volunteering
17. Unpaid work
18. Meditating
19. Private religious activities
20. Listening to music
21. Arts and crafts
22. Playing a musical instrument
23. Doing nothing
24. Other activities
25. Eating or drinking out
26. Other physical leisure
27. Schoolwork
28. Looking for a job
29. Shopping
30. Entertainment (cinema, sports, etc.)
31. Riding in a bus, trolley, car, or van
32. Visiting the beach or park
33. Visiting family or friends
34. Exercising
35. Other non-physical leisure
36. Resting
37. Attending meetings (church, AA, etc.)

| 38. | Smoking |
| 39. | Doing laundry at a laundromat |

Psychotic symptoms in EMAs were self-reported using 5 items: "Hearing voices", "Thoughts that others might want to harm you or may be untrustworthy", "Ideas that someone could read your thoughts or you could read theirs", "Receiving messages such as through the TV or radio", and "Thoughts of having special powers or abilities". Participants were asked to rate these items "since the past alarm" on a scale from 1 = "Not at all" to 7 = "Extremely". Items were summed (total range: 5–35). Adherence rates and total numbers of surveys answered were presented in the previous reports [50].

**5.3 Statistical analysis**

Analyses described below were conducted in R version 4.3.3. Participants with insufficient data (fewer than 3 cycles) were removed, and their characteristics were compared with those of remaining participants using effect sizes to consider attrition effects. For regression models, statistical significance was defined as $p < .05$ and 95% confidence intervals not overlapping the null. P-values were corrected for false discovery rate using the Benjamini & Hochberg method [52].

**5.3.1 Associations of H with momentary self-reported psychotic symptoms.** To examine associations between H and self-reported psychotic symptoms in the total sample, we used a generalized linear mixed model with observations nested in individuals. Psychotic symptoms were regressed on person-level mean H (i.e., the personal average value of H of each participant), mean-centered H (i.e., the difference between each value of H and the person-level mean H), and a random intercept. In this approach, the coefficient of person-level mean H indicates whether a tendency for higher H is associated with a tendency for more psychotic symptoms (between-person association), whereas the coefficient of mean-centered H indicates whether a person's fluctuation in H is associated with a concurrent fluctuation in psychotic symptoms (within-person association). The model was estimated using maximum likelihood with Laplace approximation and a gamma distribution with log-link function (given the skewed distribution of psychotic symptoms). A random slope of mean-centered H was added as it improved the model fit ($p < .05$) on the likelihood ratio test [53].

**5.3.2 Associations between RDMs and interviewer-rated psychotic symptoms.** To examine associations of RDMs with interviewer-rated psychotic symptoms in the schizophrenia group, we used generalized linear regression with gamma distribution and log-link function. We examined three outcomes: positive symptoms, reduced emotional experience, and reduced emotional expression. These psychotic symptom scores were regressed on each rhythmicity marker in separate models (mean H across cycles, the total H of participants' transition matrices [MH], and MEHT for each of the 9 states; thus 18 models per outcome, for a total of 54 models). Models were adjusted for site of recruitment, age, gender, ethnoracial groups, educational attainment, and mean levels of self-reported psychotic symptoms. We included the latter covariate to determine whether RDMs, the estimation of which in part relied on self-reported psychotic symptoms, could provide any additional information about interviewer-rated symptoms independently of self-reports.

**5.3.3 Group differences in RDMs.** To examine group differences in RDMs, we used linear regression. RDMs were regressed on participant groups (bipolar disorder or schizophrenia vs. control, and schizophrenia vs. bipolar disorder) in separate linear regression models for each marker. Models were adjusted for site of recruitment, age, gender, ethnoracial groups, educational attainment, and mean levels of self-reported psychotic symptoms.

**Supporting information**

**S1 Table. Comparison of participants included and excluded from analyses.** Effect sizes (range 0.00-1.00) are Cramer's V for categorical variables and absolute values of Spearman correlations for continuous variables. Effect sizes ≥ 0.10 (in bold) are considered significant.
(DOCX)

**S1 Video. (Left) Dynamic representation of the change in the transition matrix for the aggregated Markov chain in each group.** The matrix is interpolated for ease of visualization. The matrix represents the space of transitions for the Markov chains, not the statesapce of the Markov chain. (Right) Dynamic representation of the simulation results for the aggregated Markov chain in each group.
(MOV)

## Author contributions

**Conceptualization:** Axel Constant, Vincent Paquin, Robert A. Ackerman, Colin A. Depp, Raeanne C. Moore, Philip D. Harvey, Amy E Pinkham.

**Formal analysis:** Axel Constant, Vincent Paquin.

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
