## [Decision Letter · Decision Letter 0]

30 Jun 2025

Response to Reviewers
Revised Manuscript with Track Changes
Manuscript
**Journal Requirements:**

1. We ask that a manuscript source file is provided at Revision. Please upload your manuscript file as a .doc, .docx, .rtf or .tex.

2. We noticed that you used “data not shown” in the manuscript. We do not allow these references, as the PLOS data access policy requires that all data be either published with the manuscript or made available in a publicly accessible database. Please amend the supplementary material to include the referenced data or remove the references.

3. In the online submission form, you indicated that [Data are available from the authors upon reasonable request.].

a. In a public repository,

b. Within the manuscript itself, or

c. Uploaded as supplementary information.

**Additional Editor Comments (if provided):**
**Reviewers' Comments:**

**Comments to the Author**

Reviewer #1: This work explore the clinical utility of rhythmic digital markers in

schizophrenia. The study is well-founded and well-described, the data and results are clearly presented, the conclusions are fully justified. Data must be made available.

Reviewer #2: I congratulate the authors for conducting this study. The manuscript contains valuable information and contributes to the expanding body of evidence on digital markers in psychiatry. It offers hope regarding the role of digital technology in enhancing the clinical utility of mental health diagnostics and underscores the potential of digital markers in addressing psychiatric disorders. I have some observations that the authors might consider while revising their manuscript:

• I recommend abbreviating rhythmic digital markers (RDM) upon its first mention in the body of the article, specifically at line 104. “This study explores the clinical correlates of rhythmic digital markers in schizophrenia...” rather than in line 158.

• The structure and flow of the article is somewhat unclear which makes for a difficult read. Why did the authors choose to present their findings before describing the methodology? Additionally, what was the rationale behind placing future directions and limitations ahead of the study’s methodological description?

• Again, the placement of the results immediately after the introduction may be confusing for the reader. Can the authors explain why they chose to present their findings before outlining the methodology?

• I suggest that the authors include a dedicated conclusion section.

• All the best!

Reviewer #3: My main reservation regarding recommending the paper for publication concern the presentation and interpretation of the Markov chains:

1a.) 3D plots in Fig. 4 are not needed or useful. The figures are not clear, axes are not labeled, values are strangely scaled. Transition matrices should not be interpolated, they only contain values for prescribed transitions. There are no in-between values.

1b.) Values in transition cannot be interpreted as attractors or wells they are related to the solutions of the system but they describe dynamics. They do not represent the state space of the system, just the transitions between the states. The attractors/ wells in the system will be probability vectors (with probabilities of being in any of the 9 states after long time). The solutions can also have a form of (probabilistic) cycles of states.

1c.) The transition matrices are based on a very limited data (for this type of analysis); this might strongly limit reproducibility of the study

Some other comments:

2.) abstract - include n values for each group

3.) line 138 - SMS won't be clear for all audiences use also 'text'

4.) lines 160-170 - consider revising for clarity

5.) line 185 - this statement is in contradiction to the statement in lines 176 and 177 that entropy shouldn't be used; please clarify

6.) line 208, equation (1) - missing brackets around p(x)

7.) line 211 - consider revising for clarity, 'Psychologically' isn't typically used in this context. It's also a confusing explanations of entropy.

8.) Lines 217-219 - this explanation is very unclear (it kind of sounds like it says that level of confidence of in wining is 1 when it should be 0).

9.) Figure 1 - Please consider adding an illustration of constructing the activity histograms

10.) line 284 - change 'statement' to 'formula'

11.) lines 290-294 - delete; it's enough to say the you used the hittime function

12.) line 306 - example of the distribution should be normalized

13.) line 348 - MEHT is not explained

14.) lines 362-369 - delete 3d figures are not needed

15.) Figure 4. - remove the 3d panels and description in the caption

16.) line 401 - consider revising for clarity

17.) lines 404-406 - repetition

18.) lines 412, 416, 425, 430 - there is no analysis showing that discussed differences are significantly shorter/ longer

19.) lines 422-424 - I couldn't find analysis that supports that statement; please add reference to a table or figure.

20.) lines 452-456 - in my view this statement is not supported by the results in the paper

21.) lines 494-499 - consider revising for clarity

22.) lines 537-547 - please revise and tone down the statements

**Figure resubmission:****Reproducibility:** To enhance the reproducibility of your results, we recommend that authors of applicable studies deposit laboratory protocols in protocols.io, where a protocol can be assigned its own identifier (DOI) such that it can be cited independently in the future. Additionally, PLOS ONE offers an option to publish peer-reviewed clinical study protocols. Read more information on sharing protocols at https://plos.org/protocols?utm_medium=editorial-email&utm_source=authorletters&utm_campaign=protocols

---

## [Decision Letter · Decision Letter 1]

25 Aug 2025

Exploring the clinical utility of rhythmic digital markers for schizophrenia

PDIG-D-25-00164R1

Dear Dr. Constant,

We're pleased to inform you that your manuscript has been judged scientifically suitable for publication and will be formally accepted for publication once it meets all outstanding technical requirements.

Within one week, you'll receive an e-mail detailing the required amendments. When these have been addressed, you'll receive a formal acceptance letter and your manuscript will be scheduled for publication.

An invoice for payment will follow shortly after the formal acceptance. To ensure an efficient process, please log into Editorial Manager at https://www.editorialmanager.com/pdig/ click the 'Update My Information' link at the top of the page, and double check that your user information is up-to-date. For billing related questions, please contact billing support at https://plos.my.site.com/s/.

Kind regards,

Krasimira Tsaneva-Atanasova

Academic Editor

PLOS Digital Health

Additional Editor Comments (optional):

Please ensure that the outstanding minor revisions requested by Reviewer 3 are fully addressed.

Reviewers' comments:

Reviewer #3: Methodological question:

Why the authors decided to run the simulations of the MC (lines 323 to 336) rather than computing the steady state directly (for example using Matlab function asymptotics from the toolbox already used by the authors)? Direct computation would produce the same results and would simplify the methodology.

I have few final suggestion to improve clarity of presentation of the manuscript:

- Fig. 1, the bracket suggest that the authors consider transitions between c3 and c1 while they only investigate transitions between C_i and C_{i-1}

- caption of Fig. 1, at the moment it's quite unclear (it's also a bar plot not a histogram). Maybe say: "Each gray bar represents entropy (H) of the distribution of activities sampled over 3 days (up to 9 black bars representing EMAs frequencies). Each blue bar indicates the mean self-reported symptom score over the 3 days period (SC). The bar plot illustrates possible values of H and SC of a hypothetical participant over 10 three days periods/ cycles (C1-C10). [...]"

- line 296, consider changing to: "[...] possible changes (increase/ up U, decrease/ down D, no change/ flat F) in H and SC values (Figure 1 and Box 2)."

- caption of Fig. 4, say that the y axis of the histograms is in %

- line 530, instead of saying "engaged" say "transitions with probability higher than [some value] ."